# Testing for effects of growth rate on isotope trophic discrimination factors and evaluating the performance of Bayesian stable isotope mixing models experimentally: A moment of truth?

**Kirsty E. B. Gurney**[1][☯]*, **Henry L. Classen**[2], **Robert G. Clark**[1][☯]

1 Science and Technology Branch, Environment and Climate Change Canada, Saskatoon, Saskatchewan, Canada, 2 College of Agriculture and Bioresources, University of Saskatchewan, Saskatoon, Saskatchewan, Canada

☯ These authors contributed equally to this work.
* kirsty.gurney@ec.gc.ca

**Data Availability Statement:** Our entire data set is freely accessible on the Zenodo general-purpose

## Abstract

Discerning assimilated diets of wild animals using stable isotopes is well established where potential dietary items in food webs are isotopically distinct. With the advent of mixing models, and Bayesian extensions of such models (Bayesian Stable Isotope Mixing Models, BSIMMs), statistical techniques available for these efforts have been rapidly increasing. The accuracy with which BSIMMs quantify diet, however, depends on several factors including uncertainty in tissue discrimination factors (TDFs; $\Delta$) and identification of appropriate error structures. Whereas performance of BSIMMs has mostly been evaluated with simulations, here we test the efficacy of BSIMMs by raising domestic broiler chicks (*Gallus gallus domesticus*) on four isotopically distinct diets under controlled environmental conditions, ideal for evaluating factors that affect TDFs and testing how BSIMMs allocate individual birds to diets that vary in isotopic similarity. For both liver and feather tissues, $\delta^{13}C$ and $\delta^{15}N$ values differed among dietary groups. $\Delta^{13}C$ of liver, but not feather, was negatively related to the rate at which individuals gained body mass. For $\Delta^{15}N$, we identified effects of dietary group, sex, and tissue type, as well as an interaction between sex and tissue type, with females having higher liver $\Delta^{15}N$ relative to males. For both tissues, BSIMMs allocated most chicks to correct dietary groups, especially for models using combined TDFs rather than diet-specific TDFs, and those applying a multiplicative error structure. These findings provide new information on how biological processes affect TDFs and confirm that adequately accounting for variability in consumer isotopes is necessary to optimize performance of BSIMMs. Moreover, results demonstrate experimentally that these models reliably characterize consumed diets when appropriately parameterized.

open repository (https://doi.org/10.5281/zenodo.10927544).

**Funding:** Funding was provided by Environment and Climate Change Canada. The funders had no role in study design, data collection and analysis, decision to publish, or preparation of the manuscript.

**Competing interests:** The authors have declared that no competing interests exist.

# Introduction

In applications related to trophic ecology, stable isotopes (i.e., the naturally occurring ratio of heavy to light isotopes, primarily for carbon, $\delta^{13}$C, and nitrogen, $\delta^{15}$N) are common chemical tracers [1, 2]. Stable isotope analyses are particularly well established for diet reconstruction and expand the scope of inference that can be obtained from other dietary methods, particularly in systems where direct observations are not possible and potential food items are isotopically distinct [3–5]. When metabolically inert tissues such as feathers, hair, or nails are sampled, such methods offer the additional advantage of being non-lethal and less invasive than sampling of other tissues, like blood or muscle [6, 7]. To estimate the proportional contributions of different food resources to consumer tissues, stable isotope analyses primarily use Bayesian stable isotope mixing models (BSIMMs) [8, 9].

Broadly, by using a variety of biological and chemical tracers, statistical models of mixtures (i.e., mixing models) quantify the relative contributions of different source materials to composite, end-member samples [10–12]. Given mathematical limitations of simple linear mixing models, however, BSIMMs, which are flexible and based on a likelihood structure, are most commonly used across the natural sciences to identify proportional inputs of sources to a mixture [13, 14]. Such models incorporate variability into model inputs, allow for inputs from multiple possible sources, and generate potential end-mixture solutions as true probability distributions [15]. Importantly, the Bayesian approach improves upon frequentist methods because processes known to affect the variability in the mixture data are accounted for in model error structures, so that probability distributions reflect associated uncertainties and allow inference about which solutions are more likely than others [16, 17].

The accuracy with which BSIMMs identify source contributions to consumer tissues is strongly influenced by model specifications, especially error assumptions and variation in consumer and resource isotope values related to a suite of factors that can often not be controlled or accounted for [16, 18]. In addition, stable isotope ratios of food items change when incorporated into the tissues of consumers–these changes are described by a trophic discrimination factor (TDF), a key input to BSIMMs that further affects model outputs [19, 20]. Most commonly derived from feeding trials of captive animals, TDFs are highly variable and generally depend on species, diet, and tissue type, although our understanding of how different factors affect TDFs, particularly for carbon, remains limited [21, 22].

To date, most evaluations of how accurately BSIMMs predict diet have been conducted through simulations [16], or controlled studies of captive [19, 23] or wild [5, 24] animals, and reach different conclusions about the relative importance of TDFs on bias, precision, and accuracy of posterior estimates, as well as the factors that affect TDFs. For example, some studies suggest that species-specific TDFs are of secondary importance to the use of informative priors (i.e., existing knowledge about dietary composition) [25], whereas others find that use of an appropriate TDF has a stronger effect on performance of BSIMMs [19, 23, 24]. Further, some data indicate that TDFs vary not only based on nutrient composition of consumed foods and consumer type (i.e., omnivorous, carnivorous), but also on interactions of these factors–and that the factors affecting TDFs vary for different isotopes [22, 26]. An experimental study with tilapia (*Oreochromis mossambicus*) showed the importance of diet type and tissue type in determining TDFs, with simulated data providing additional evidence that the use of experimentally-derived TDFs, compared to those from the literature, improved accuracy of dietary estimates [19]. This is similar to Stephens et al.'s (2022) findings that using existing data and phylogenetic relatedness (i.e., Stable Isotope Discrimination Estimation in R; SIDER) [27] to derive TDFs can lead to misleading conclusions about diet, but is in contrast to other studies that suggest such methods improve BSIMMs performance, or that TDFs affect model results

less than other factors [28, 29]. Reliable information about factors affecting TDFs is needed to resolve these apparently contrasting results and to more clearly understand the true, best-case accuracy of BSIMMs outputs in trophic ecology, as well as in other applications [30, 31].

To identify the most critical factors to consider in applying BSIMMs and thus improve inference related estimates of animal diet, we aim to test for factors that affect TDFs and to evaluate how different sources of variation in isotope values drive uncertainty in BSIMM outputs. We test the performance of BSIMMs experimentally, by feeding domestic broiler chicks (*Gallus gallus domesticus*) isotopically distinct diets under controlled environmental conditions, asking three key research questions: (i) how does variation among individual growth rates affect TDFs for carbon ($\Delta^{13}$C) and nitrogen ($\Delta^{15}$N) in an omnivorous bird; (ii) how do variable estimates of TDFs and different estimates of uncertainty in BSIMMs influence model performance; and (iii) what is the best-case accuracy of BSIMM outputs when environmental and resource-related properties are primarily controlled for? Although broiler chickens have undergone extensive artificial selection for growth in captivity, variation in growth rate remains [32]. To our knowledge, although the effects of individual growth rates on TDFs have been demonstrated in fish species, similar evaluations have not been conducted in endotherms with higher basal metabolic rates, like birds [33, 34], but see [35]. Because variation in tissue isotopic values related to environmental factors and dietary composition (isotopic values and source of macronutrients) is minimized in our captive setting, the broiler chicks are an ideal system to test how variation in growth and other organism-dependent factors, such as tissue type, affect TDFs and how accurately BSIMMs allocate individual birds to assigned dietary groups when model inputs vary.

We anticipated that both individual growth rate and tissue type would be important predictors of $\Delta^{13}$C values, with $\Delta^{13}$C decreasing as growth rate increases and routing of dietary constituents (especially amino acids) to tissues is prioritized over the production of energy (the "anabolism hypothesis" [36, 37]. We also expected the growth rate effect to be tissue dependent, being more evident in liver, a tissue that is vital for bodily function and has more rapid turnover, relative to feathers, which are metabolically inert [38]. For $\Delta^{15}$N, we expected that tissue type would have a similar effect as $\Delta^{13}$C (liver having lower values compared to feathers), but that effects of metabolic routing would be reduced, resulting in a lesser effect of growth on $\Delta^{15}$N, relative to $\Delta^{13}$C [39]. Model outputs from BSIMMs should provide the most accurate assessments of assimilated diets when observed variation in TDFs (i.e., diet-specific $\Delta^{13}$C and $\Delta^{15}$N values) are applied [19, 40]. In terms of uncertainty estimates, we presume that birds in our study could not preferentially consume dietary components, so that process error (i.e., individual specialization) will have a minimal influence on isotopic variation of tissues. We therefore predict that models specifying a residual error-only formulation will be most supported by the data [15, 16].

## Methods

### Diet composition and experimental design

Using commercially available feed ingredients, we designed two isotopically distinct, nutritionally balanced diets to meet or exceed nutrient specifications for as-hatched Ross-308 broiler chickens [41]. Detailed summaries of dietary constituents are available as Supporting Information (S1 Table). The two primary diet formulations (diet 1 and diet 2) were manufactured in mash form, with two secondary formulations based on mixtures: diet 3 (67% diet 1; 33% diet 2) and diet 4 (33% diet 1; 67% diet 2). All four diets were then pelleted (< 80°C in a conditioner) and crumbled, with each dietary treatment designated as an experimental group. Day of hatch, mixed-sex Ross-308 broiler chicks were purchased from a commercial hatchery

(Sofina Foods Inc., Wynyard, Saskatchewan, Canada), weighed, individually banded, and randomly assigned to one of 24 cages (six replicate cages per group). Each experimental unit (i.e., cage) contained four chicks, and to replace any birds that died during the first week of the trial, we also maintained extra chicks for each group (n = 10 per group) in separate cages.

Housing procedures followed standard production guidelines for caged broiler chickens, and we obtained written consent and approval for all protocols from the Animal Care Committee (Animal Research Ethics Board), University of Saskatchewan, on behalf of the Canadian Council on Animal Care (Protocol 19940248, Poultry Centre File Number CTR 1702). We placed individual birds in the top compartments (51 cm W x 51 cm L x 46 cm H; 650 cm$^2$ per bird) of two-tier battery cages and gave birds *ad libitum* access to feed (front-mounted feed trough) and water (nipple, automatic, adjustable height). Extra feed was provided in supplemental trays for the first four days, and water was provided in ice cube trays for the first week. Room temperature was gradually reduced (3˚C weekly) from 34˚C to 22˚C. The light:dark cycle began at 23:1 for day-old chicks and was gradually adjusted to 19:5 by day six, where it remained for the remainder of the trial (total of 29 days). The light period included 15-minute dawn and dusk periods, and light intensity at the feeder was 20 lux. To eliminate the possibility of dietary contamination across groups, chicks were placed only in the top tier of cages, and an empty cage was left between experimental replicates.

To index feed intake and growth rates, changes in food mass were estimated over a 24-hour period throughout the trial and on days one, 14, and 29, remaining chicks (n = 93; one bird censored due to a missing mass value at day 29) were weighed on a top-loading balance (± 1 g). Dietary samples from each group were collected throughout the trial (weekly) to test isotope values of feed. After 29 days, we euthanized all birds via cervical dislocation. We immediately excised the entire liver, collected flank-breast feathers, and sexed birds via visual confirmation of testes or ovaries. Liver and feather samples were stored in sterile polyethylene bags at -80˚C (liver) or in paper envelopes at -18˚C (feathers).

## Stable isotope analyses

Feather samples were rinsed with a 2:1 chloroform-methanol solution, allowed to air dry for 24 hours, and finely cut into pieces (calamus removed); liver samples were freeze-dried and ground to a homogenous powder; and diet samples were finely powdered in a homogenizer. To remove lipids, liver and diet samples were rinsed with a 2:1 chloroform-methanol solution and dried to constant mass. All tissue and diet samples were packed (approximately 1 mg) into tin capsules and combusted in a Costech (Milan, Italy) ECS4010 elemental analyzer interfaced with a Delta V continuous-flow isotope-ratio mass spectrometer at the Department of Soil Science, University of Saskatchewan. By convention, isotope values are reported in delta notation as $\delta^{13}$C and $\delta^{15}$N according to:

$$\delta X = \left( \frac{R_{sample}}{R_{std}} - 1 \right)\%$$

where X = $^{13}$C or $^{15}$N, R = $^{13}$C:$^{12}$C or $^{15}$N:$^{14}$N, and international standards (std) are Vienna PeeDee Belemnite (VPDB) for $^{13}$C and atmospheric N$_2$ (AIR) for $^{15}$N, and where measured delta values were converted to international isotope reference scales using two-point linear normalization [42].

Two secondary isotopic reference materials (egg albumen and lyophilized bowhead whale baleen) that were expected to span the range of experimental values were selected by the analytical lab, as standards to allow estimation of measurement precision [43]. Standards–provided by the lab (in the same mass range as test samples)–were calibrated by running them

against USGS 40 and USGS 41a L-glutamic acids, which have internationally accepted $\delta^{13}$C and $\delta^{15}$N values [44, 45]. For further quality assurance / quality control, standards were processed before and between every five tissue samples within each analytical run (for albumen) and at the start, midpoint, and end of each run (for baleen). We used replicate standard measurements within a run to estimate analytical precision: mean $\delta^{13}$C (± 1 standard deviation, SD) within runs = –22.4 ‰ ± 0.0 ‰, and mean $\delta^{15}$N (± 1 SD) within runs = 6.8 ‰ ± 0.1 ‰ for egg albumen; mean $\delta^{13}$C (± 1 SD) within runs = –18.5‰ ± 0.1%, and mean $\delta^{15}$N (± 1 SD) within runs = 14.8 ‰ ± 0.2%, for bowhead whale baleen. Measurement error estimates (± 1 SD) were ± 0.1 ‰ for $\delta^{13}$C and ± 0.3 ‰ for $\delta^{15}$N, based on measurements of laboratory standards across multiple runs [43].

## Statistical approach

To evaluate the relative influence of organism-related factors (growth rate and tissue type) on TDFs, analyses were implemented in SAS® Version 9.4 (SAS® Inc., Cary, NC). We calculated $\Delta^{13}$C as $\delta^{13}$C$_{Tissue}$– $\delta^{13}$C$_{Diet}$ and $\Delta^{15}$N as $\delta^{15}$N$_{Tissue}$– $\delta^{15}$N$_{Diet}$ for both liver and feathers, estimated growth parameters for each individual, based on three-parameter Gompertz growth curves [46, 47], and used general linear mixed models for our analyses. These models allowed us to account for dependence among birds at the cage level (random effects), while evaluating the specific and separate influences of our predictor variables of interest [48]. For each TDF, model covariance structure was determined based on data structure and convergence criteria [49]. Marginal log-likelihood values for each model were computed using restricted maximum likelihood methods, and a Kenward-Roger correction was applied in computing denominator degrees of freedom (PROC MIXED) [50]. After developing an *a priori* set of candidate models (one each for $\Delta^{13}$C and $\Delta^{15}$N), we compared models based on an information-theoretic approach, with models ranked according to 2nd-order Akaike's Information Criterion corrected for sample size (AIC$_c$) and relative likelihoods for each model estimated by Akaike weights ($w_i$) [51].

Key predictor variables (fixed effects) in our models included individual growth rate based on mass (Mass$_B$), tissue type, and their interaction (Mass$_B$*Tissue). To account for additional factors known to affect TDFs, we also included diet type (Group) and sex in our model sets [40, 52] and evaluated potential interactions between diet and tissue (Group*Tissue) and sex and tissue (Sex*Tissue). Inference concerning fixed effects was based on precision (85% confidence interval, CI) of the regression coefficient (β), as this interval is more compatible with the information-theoretic approach. When wide confidence intervals indicated imprecise slope estimates, we considered respective covariates as uninformative [53].

To compare the effects of different TDFs and error structures on stable isotope dietary inference and test how accurately BSIMMs estimated known proportions of the diets that were consumed, we used hierarchical BSIMMs, implemented in the open-source R package, MixSIAR Version 3.1.10 [54]. Within MixSIAR, each model was first run under test settings to confirm models were correctly specified–cage as a random effect, nested within the fixed effect of group (Cage(Group)). For final analyses, Gibbs sampling parameters were set to 3 parallel chains of 300,000 vectors (burn-in = 200,000, thinning interval = 100). To determine whether mixing models converged, we examined trace plots for relatively constant mean and variance, as well as for small fluctuations in the estimated parameters–results from the Gelman-Rubin diagnostic test were also considered [54]. TDFs were specified as diet-specific (mean ± standard deviation)(Diet 1: $\Delta^{13}$C = 0.7 ± 0.2, $\Delta^{15}$N = 0.7 ± 0.2; Diet 2: $\Delta^{13}$C = 2.3 ± 0.2, $\Delta^{15}$N = 1.2 ± 0.4) or combined ($\Delta^{13}$C = 1.4 ± 0.8, $\Delta^{15}$N = 0.9 ± 0.3). Different specifications of BSIMMs that converged were ranked using the Deviance Information

Criterion, DIC [55], and to estimate the relative accuracy of model outputs across groups, we compared the actual diet proportions to the model-based 95% credible intervals–when intervals included the known proportion, we considered the output accurate.

## Results

### Diet composition and chicken growth

Feed isotopic values ($\delta^{13}$C, $\delta^{15}$N) were significantly different among the experimental diet groups (MANOVA, Wilks' $\lambda = 0.007$, $F_{6, 54} = 97.06$, $p < 0.0001$); as expected, our formulations resulted in a high $\delta^{13}$C / $\delta^{15}$N diet (Group 1), a low $\delta^{13}$C / $\delta^{15}$N diet (Group 2) and 2 intermediates (Groups 3 and 4) (Fig 1).

On day one, chick body mass averaged 47.1 g (± 1.0 g standard error, SE), and no effects of sex or dietary grouping were detected. Group 3 had three more females than males and Group 1 had six more males than females, but sex ratio did not differ among groups (G-test, $G_3 = 1.91$, $p = 0.59$); overall, there were 46 females and 48 males (Table 1). By day 29, after accounting for cage effects, additional effects of sex ($p < 0.0001$) and diet group ($p = 0.01$) on body mass were detected, with no interaction. Across all birds, coefficients for daily rate of mass gain ($Mass_B$) ranged from 0.0403 (a bird in Group 3) to 0.0907 (a bird in Group 3), solving the Gompertz equations for days 15 and 16, estimated values were equivalent to 34 to 83 g of body mass gained per day. Although we detected no effects of sex or diet group (or their interaction) on $Mass_B$, by the end of the diet trial, female chickens (least squares mean = 1621 g ± 28 SE) averaged ~144 g less than males (1765 g ± 28 SE). Adjusting for sex effects, least square mean body masses (SE) of chickens were 1732 g (46), 1737 g (46), 1539 g (47), 1736 g (46) for diet Groups 1–4, respectively.

### Stable isotope values & factors affecting TDFs

For both liver and feather tissue, $\delta^{13}$C and $\delta^{15}$N values differed among dietary groups, reflecting patterns in feed isotope values; compared to feather, $\delta^{13}$C was generally more negative for liver, with the opposite pattern for $\delta^{15}$N (Fig 2). Similarly, as expected, $\Delta^{13}$C was affected by dietary group, tissue type, and their interaction, but models including an effect of sex on $\Delta^{13}$C were not supported. Consistent with our predictions under the anabolism hypothesis, we detected a negative effect of $Mass_B$ on $\Delta^{13}$C, for liver tissue only ($\beta_{MassB*Feather} = 13.7$, 85% CI = 7.0 to 20.5), with some minor differences among groups (Table 2, Fig 3). For $\Delta^{15}$N, the most-supported model included effects of dietary group, sex, and tissue type, as well as an interaction between sex and tissue type ($w_i = 0.63$; $\beta_{Female*Feather} = -1.30$, 85% CI = -0.19 to -0.06), although a model without the interaction also received moderate support ($w_i = 0.36$, Table 3). The sex effect (females having higher $\Delta^{15}$N relative to males) was observed only for liver, and contrary to our expectation, in general, $\Delta^{15}$N values were greater for liver than feather (Fig 4).

### Influences of TDFs & error structure on performance of BSIMMs

For liver tissue, our top-ranked BSIMM (based on DIC) showed that combining the variance in TDFs across diet types and applying a multiplicative error structure would improve model accuracy compared to the residual error-only model ($\Delta$DIC = 6.6; Table 4). Models with diet-specific TDF values for $\Delta^{13}$C and $\Delta^{15}$N either did not converge (multiplicative error) or were poorly supported by the data (residual error). Results for feather models were similar in that increasing variance estimates for TDF values–by combining diet-specific values–and using a multiplicative error structure resulted in the most-supported model, although there was some

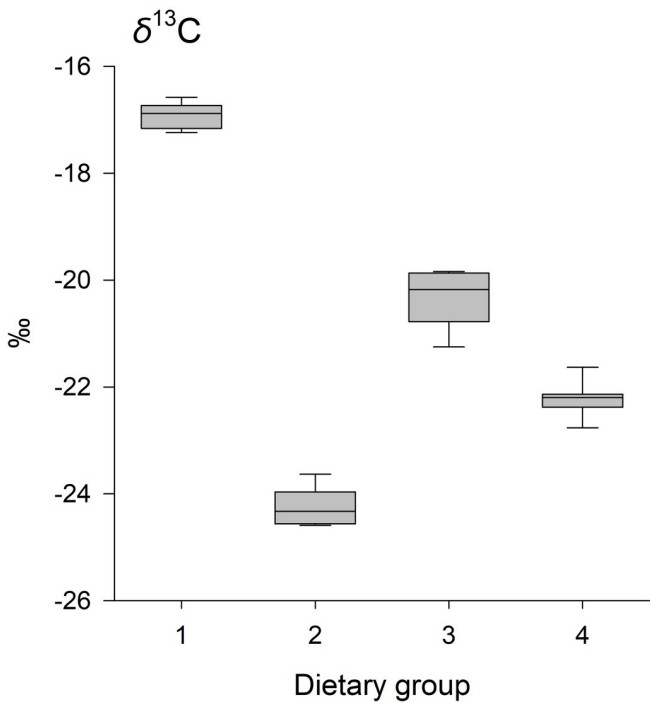

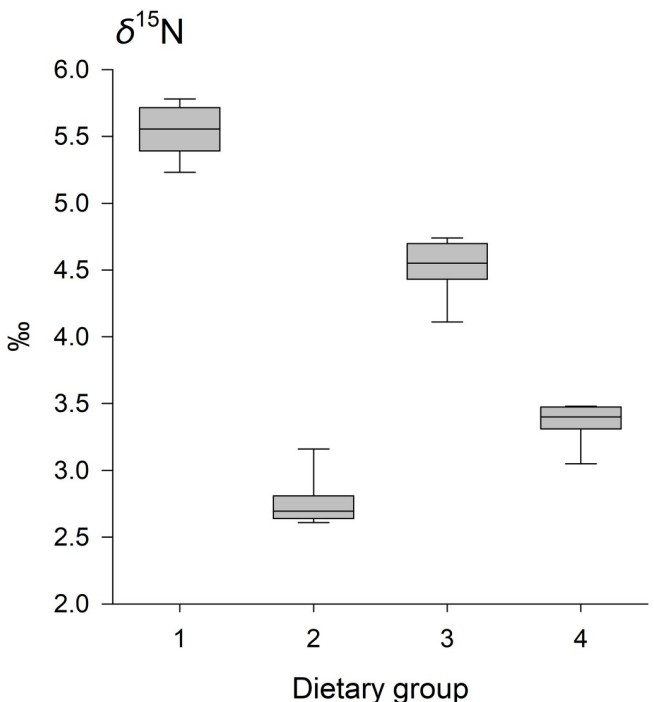

**Fig 1. Dietary $\delta^{13}$C (upper panel) and $\delta^{15}$N (lower panel) by experimental dietary group.** Isotope values differed among diets fed to growing Ross-308 strain chicks under controlled environmental conditions. Shown are the median (horizontal line within shaded boxes), lower and upper quantiles (lower and upper edge of box, respectively) and range (vertical line) for each dietary group (n = 8 samples per group).

**Table 1. Descriptors of growth for Ross-308 broiler chicks.**

| Dietary Group | Sex | n | Mass$_{29}$ (g) | B | M$_m$ (g) | t* |
|---|---|---|---|---|---|---|
| 1 | Female | 9 | 1655 (33) | 0.0634 (0.0015) | 3510 (181) | 24 (1) |
| | Male | 15 | 1807 (28) | 0.0639 (0.0018) | 3915 (218) | 25 (1) |
| 2 | Female | 12 | 1665 (38) | 0.0652 (0.0013) | 3362 (157) | 23 (1) |
| | Male | 12 | 1809 (56) | 0.0626 (0.0029) | 4289 (414) | 26 (1) |
| 3 | Female | 13 | 1426 (66) | 0.0576 (0.0022) | 3382 (149) | 27 (1) |
| | Male | 9 | 1657 (76) | 0.0625 (0.0040) | 3800 (344) | 25 (2) |
| 4 | Female | 12 | 1704 (26) | 0.0626 (0.0017) | 3704 (199) | 25 (1) |
| | Male | 11 | 1819 (41) | 0.0611 (0.0017) | 4156 (175) | 26 (1) |

Mean (standard error) for each dietary group were estimated using a Gompertz three-parameter equation, $M_t = M_m \exp(-\exp(-B(t-t^*)))$, where $M_t$ = mass at time $t$, $M_m$ = estimated mass at maturity, B = coefficient for daily rate of mass gain, and t* = day on which mass gain is maximized.

indication that a residual error-only model (with combined TDF variance) performed well (ΔDIC = 2.5; Table 5). Outputs from our top-ranked BSIMMs were generally very similar to known dietary contributions, with some variation in accuracy, depending on group and tissue type. For liver, estimated dietary proportions based on 95% credible intervals were accurate (i.e., included known proportions) for dietary groups 3 and 4, whereas for feather–best model outputs were accurate for group 4 only (Tables 4 and 5). Elsewise, credible intervals generally were within 10% of known proportions (Fig 5).

## Discussion

Using captive, domestic birds being fed specifically formulated, isotopically distinct diets, we have directly measured how several factors, including diet, tissue type, sex, and individual growth rates, affect trophic discrimination factors and, further, how variation in the specifica-tions of Bayesian stable isotope mixing models affect the accuracy of their outputs. Although TDFs varied considerably across dietary groups and tissue types (as expected), our work high-lights two other factors that influence TDFs in birds–sex and individual growth rate. Impor-tantly, we also show that adequate estimates of uncertainty in BSIMMs improve performance of these models, and that when variation in the isotope values of animals due to environment and food resources is controlled (i.e., secondary mechanistic factors; 18), BSIMM-derived esti-mates of consumed diet can closely reflect known proportions.

### Factors influencing trophic discrimination factors

As expected, our analyses indicated a negative relationship between individual growth rate and $\Delta^{13}$C –for liver tissue only. Although the relationship between growth rates and TDFs has rarely been investigated directly, this finding is generally aligned with other studies of verte-brates that suggest an important role for metabolic processes in determining trophic discrimi-nation factors [56–58]. For example, faster growing juvenile fish tend to have higher metabolic rates, relative to slower-growing conspecifics, which in turn is linked to decreases in $\Delta^{13}$C [36, 59]. It is unclear if adaptations to endothermy would result in similar relationships for birds. Intraspecific variation in percentage mass change did not affect turnover rates of $^{13}$C in dunlin (*Calidris alpina*), and $\Delta^{13}$C did not differ predictably between slower and faster-growing cap-tive American crows (*Corvus brachyrhynchos*) [38, 60]. Yet $\Delta^{13}$C in domestic chickens ranges from approximately -4‰ to 2‰, and observed variation among captive-raised *Caipirinha* broilers suggests a growth effect in this species [40, 61], as do the lower TDFs for younger age

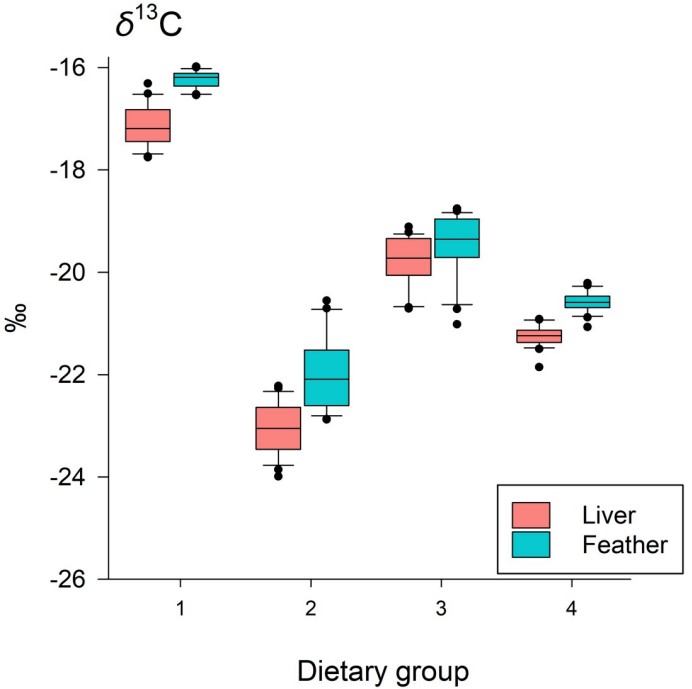

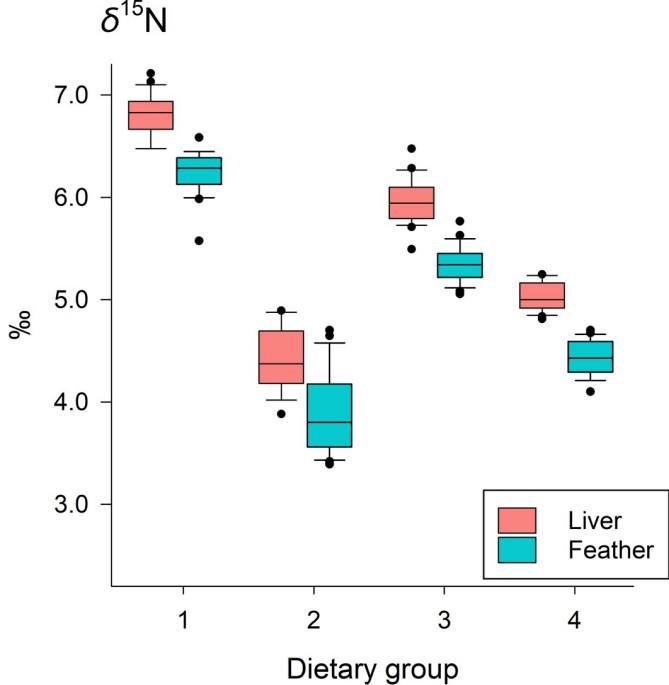

**Fig 2. Tissue $\delta^{13}$C (upper panel) and $\delta^{15}$N (lower panel) for liver and feather.** Values varied across dietary groups. Shown are the median (horizontal line within boxes), lower and upper quantiles (lower and upper edge of box, respectively) and range (vertical line) for each dietary group, with values above or below the interquartile range (upper quartile–lower quartile) indicated by closed circles.

**Table 2. Model ranking for $\Delta^{13}$C.**

| Model Structure | K | -2logL | $\Delta AIC_c$ | $w_i$ |
|---|---|---|---|---|
| Group*Tissue + Mass$_B$*Tissue | 12 | 64.7 | 0.00 | 1.0 |
| Group*Tissue + Mass$_B$ | 11 | 78.0 | 11.01 | 0.0 |
| Group + Tissue + Mass$_B$ | 8 | 132.1 | 58.41 | 0.0 |
| Group + Tissue | 7 | 139.4 | 63.53 | 0.0 |
| Intercept | 3 | 329.4 | 245.03 | 0.0 |

Results are consistent with tissue-dependent effects of individual growth rate (Mass$_B$) on $\Delta^{13}$C. For all models, cage nested within group is included as a random effect. -2logL = Deviance, $w_i$ = Akaike weight. K = number of parameters estimated. The + between variables indicates an additive effect, the * denotes interaction; where interactions are listed, main effects were also included.

classes that have been observed in other species [62]. Our results thus add to a growing body of evidence that differences in physiology among individuals have an important influence on TDFs, which we speculate is linked to increased routing of dietary protein to body tissues during rapid growth (i.e., anabolism). That we observed an inverse relationship between growth rate and $\Delta^{13}$C only in liver, a highly metabolically active tissue (relative to feathers), is consistent with this idea [63, 64]. However, differences in the specific amino acid composition of livers and feather (i.e., keratin) may be an additional factor affecting differences in fractionation values between tissue types and the interaction with growth rate [65, 66].

For $\Delta^{15}$N, our finding that diet and tissue type are reliable predictors of fractionation values is consistent with expectations, which were based on other studies that consistently highlight the importance of these factors, especially dietary composition [21, 22, 67]. The sex dependence of the tissue effect on $\Delta^{15}$N, however, was not anticipated. In liver only, fractionation values for nitrogen were marginally higher for female chicks than for males. Although few studies have specifically evaluated sex effects on TDFs, higher $\Delta^{15}$N in females have previously been noted in some mammalian species [68, 69]. Although sex-based variation in growth is a proposed mechanistic explanation in both cases, this hypothesis is unsupported by our study, where there was no evidence of a growth rate effect on $\Delta^{15}$N. Instead, physiological differences related to oogenesis may play an important role. If female broiler chicks increase routing of nitrogen to liver for production of eggs–even prior to sexual maturity–enrichment of $^{15}$N would be expected [70].

We did not design our feeding trial to specifically evaluate the influence of dietary composition (isotopic values and sources of macronutrients) on trophic discrimination factors for carbon or nitrogen, but our findings that dietary group affected both $\Delta^{13}$C and $\Delta^{15}$N were not unexpected. Accounting for other factors, in our study, $\Delta^{13}$C and $\Delta^{15}$N were generally highest for dietary groups 2 and 4 (Figs 3 and 4), which had reduced $\delta^{13}$C values (Fig 1), a greater percentage of $C_3$-based carbon (i.e., wheat and soy), and less fish meal, relative to Groups 1 and 3. Although evidence of an influence of $\delta^{13}$C values on $\Delta^{13}$C in birds is lacking [71], origin of carbon sources (i.e., $C_3$-based, $C_4$-based, marine-based, or a mixture) and dietary protein content can explain substantial variation in $\Delta^{13}$C for avian species [39, 40]. Given similar estimated protein content across the different diets in our study, our data support a carbon source hypothesis, but we note that differences in digestibility of different protein sources may also be important, not only for $\Delta^{13}$C, but also for $\Delta^{15}$N [67, 72]. In particular, for $\Delta^{15}$N, our results are consistent with an important influence of protein quality; our diet groups with higher levels of easily-digestible fish protein also had lower $\Delta^{15}$N values. Additional studies–especially those

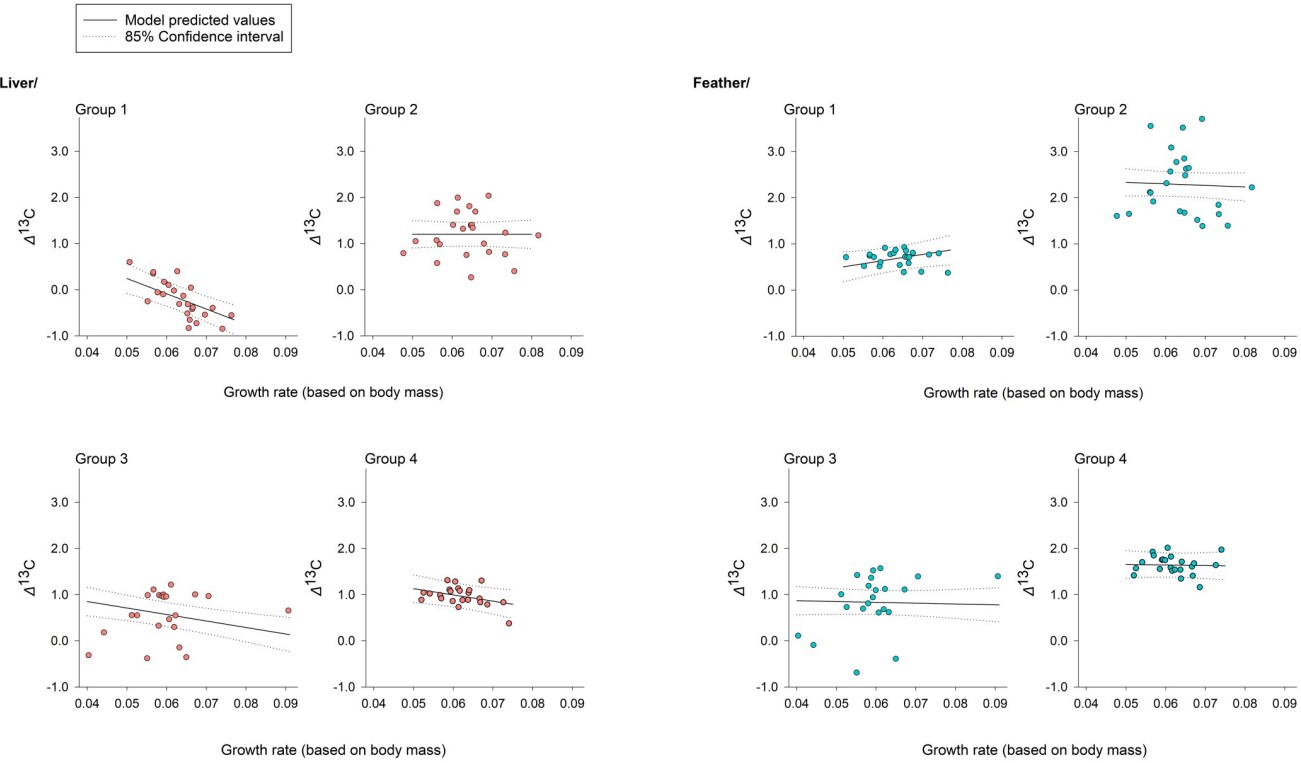

**Fig 3. $\Delta^{13}$C in captive broiler chicks was affected by individual growth rate.** The effect varied across dietary groups and between tissue types. All plots show model-based estimates (± 85% confidence intervals) as indicated in the legend. Raw data (circle symbols) are plotted for visual comparison.

using specifically formulated food sources and compound-specific isotope analysis of amino acids–will provide improved insight about how diet composition affects $\Delta^{13}$C [73, 74].

## Performance of Bayesian stable isotope mixing models

Varying TDF estimates and error structures in our BSIMM formulations affected performance of the models (based on DIC and estimated dietary contributions), with variance in TDFs having the greater effect, both for liver and feather tissue. Although we anticipated that model performance would be improved by specifying diet-specific $\Delta^{13}$C and $\Delta^{15}$N values–we observed

**Table 3. Model ranking for $\Delta^{15}$N.**

| Model Structure | K | -2logL | $\Delta$AIC$_c$ | $w_i$ |
|---|---|---|---|---|
| Group + Sex*Tissue | 9 | -84.5 | 0.00 | 0.63 |
| Group + Tissue + Sex | 8 | -81.2 | 1.09 | 0.36 |
| Group + Tissue | 7 | -72.3 | 7.81 | 0.01 |
| Intercept | 3 | 170.0 | 241.61 | 0.00 |

The model most supported by our data identified that sex affected $\Delta^{15}$N for liver tissue only. Cage nested within group is included as a random effect. -2logL = Deviance, $w_i$ = Akaike weight. K = number of parameters estimated. The + between variables indicates an additive effect, the * denotes interaction; where interactions are listed, main effects were also included.

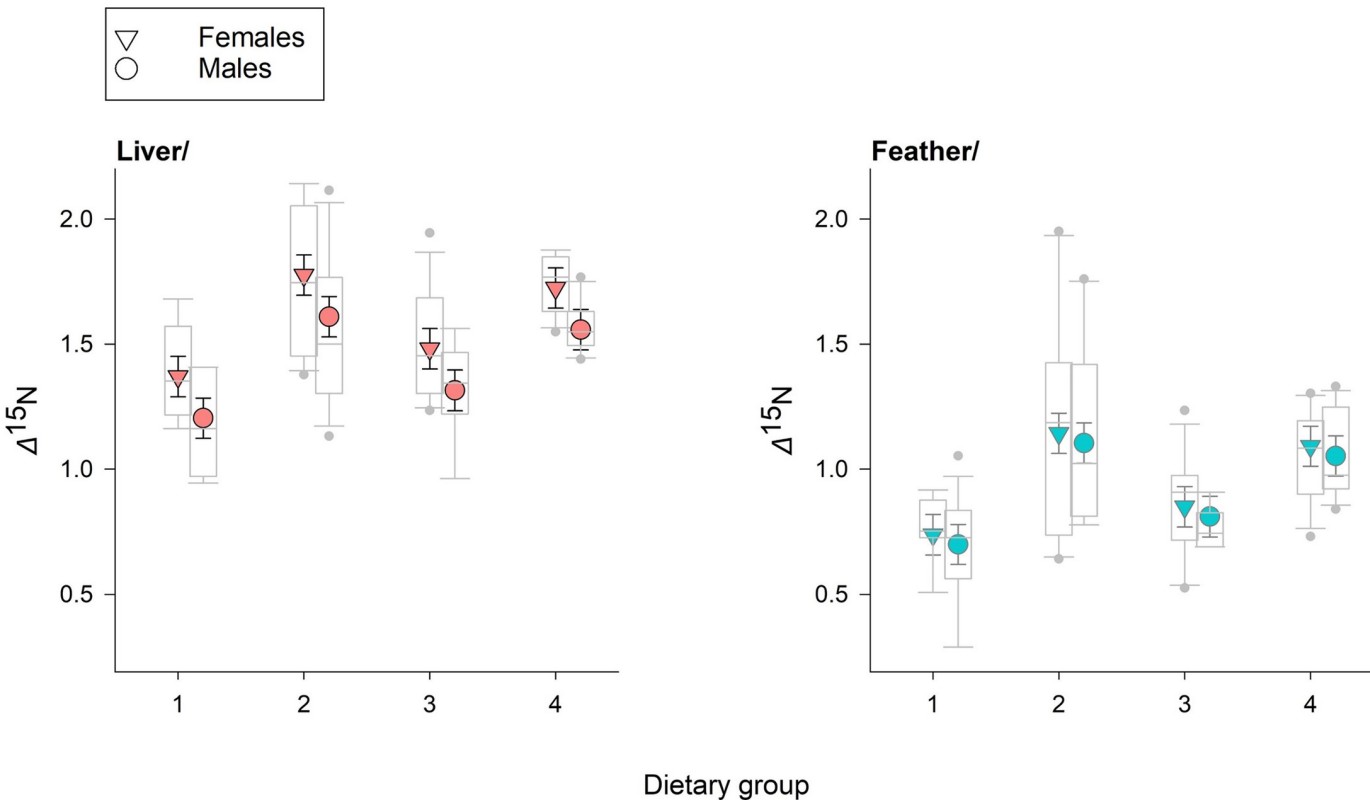

**Fig 4. Dietary group, sex, and tissue type influenced $\Delta^{15}$N in captive broiler chicks.** For liver, $\Delta^{15}$N was higher in females (triangles), but for feathers, values were similar between females and males (circles). All plots show model-based estimates (± 1 standard error). Raw data (grey boxplots) are plotted for visual comparison.

**Table 4. Bayesian stable isotope mixing model outputs for liver tissue.**

| TDF-Variance | Error term | Dietary group 1 | | Dietary group 2 | | Dietary group 3 | | Dietary group 4 | | DIC |
|---|---|---|---|---|---|---|---|---|---|---|
| | | Formula 1 (1)[a] | Formula 2 (0) | Formula 1 (0) | Formula 2 (1) | Formula 1 (0.67) | Formula 2 (0.33) | Formula 1 (0.33) | Formula 2 (0.67) | |
| Combined | P*R | 0.87, 0.78–0.91 | 0.13, 0.09–0.22 | 0.09, 0.05–0.14 | 0.91, 0.86–0.95 | 0.56, 0.45–0.67 | 0.44, 0.33–0.55 | 0.33, 0.25–0.44 | 0.67, 0.56–0.76 | 38.8 |
| Combined | R | 0.87, 0.80–0.91 | 0.13, 0.09–0.20 | 0.08, 0.05–0.13 | 0.92, 0.87–0.95 | 0.55, 0.45–0.66 | 0.45, 0.35–0.56 | 0.34, 0.24–0.44 | 0.66, 0.56–0.76 | 45.4 |
| Diet specific | P*R | Did not converge | | | | | | | | |
| Diet specific | R | 0.87, 0.49–0.96 | 0.13, 0.04–0.51 | 0.06, 0.02–0.26 | 0.94, 0.74–0.98 | 0.61, 0.30–0.87 | 0.39, 0.13–0.70 | 0.39, 0.16–0.76 | 0.61, 0.24–0.84 | 117.4 |

Using trophic discrimination factors with increased variance (i.e., combined across diet types) and an error structure that allows for combinations of process error (i.e., individual specialization and sampling error) as well as residual error, yielded the best-approximating Bayesian stable isotope mixing model. Model specifications are identified in the first two columns, known dietary proportions are shown for each experimental group[a], as well as estimated dietary contributions from each model (median, 95% credible interval). Where the credible interval includes the known proportion (i.e., model output is accurate), cells are shaded green. P = process error, R = residual error. DIC = Deviance Information Criterion.

[a] Bolded values in parentheses indicate known dietary proportions.

**Table 5. Bayesian stable isotope mixing model outputs for feather tissue.**

| TDF-Variance | Error term | Dietary group 1 | | Dietary group 2 | | Dietary group 3 | | Dietary group 4 | | DIC |
|---|---|---|---|---|---|---|---|---|---|---|
| | | Formula 1 (1)[a] | Formula 2 (0) | Formula 1 (0) | Formula 2 (1) | Formula 1 (0.67) | Formula 2 (0.33) | Formula 1 (0.33) | Formula 2 (0.67) | |
| Combined | P*R | 0.90, 0.84–0.93 | 0.10, 0.07–0.16 | 0.13, 0.08–0.19 | 0.87, 0.81–0.92 | 0.49, 0.39–0.59 | 0.51, 0.41–0.61 | 0.33, 0.25–0.43 | 0.67, 0.57–0.75 | 66.8 |
| Combined | R | 0.89, 0.81–0.93 | 0.11, 0.07–0.19 | 0.12, 0.07–0.18 | 0.88, 0.82–0.93 | 0.51, 0.40–0.62 | 0.49, 0.38–0.60 | 0.33, 0.24–0.45 | 0.67, 0.55–0.76 | 69.3 |
| Diet specific | P*R | 0.84, 0.19–0.97 | 0.16, 0.04–0.81 | 0.10, 0.02–0.54 | 0.90, 0.49–0.98 | 0.56, 0.11–0.85 | 0.45, 0.15–0.89 | 0.39, 0.11–0.82 | 0.61, 0.18–0.89 | 91.7 |
| Diet specific | R | 0.83, 0.28–0.95 | 0.17, 0.05–0.72 | 0.11, 0.02–0.55 | 0.89, 0.45–0.98 | 0.60, 0.22–0.89 | 0.40, 0.11–0.79 | 0.40, 0.10–0.79 | 0.60, 0.21–0.90 | 108.8 |

Using trophic discrimination factors with increased variance (i.e., combined across diet types) and an error structure that allows for combinations of process error (i.e., individual specialization and sampling error) as well as residual error, yielded the best-approximating Bayesian stable isotope mixing model. Model specifications are identified in the first two columns, known dietary proportions are shown for each experimental group[a], as well as estimated dietary contributions from each model (median, 95% credible interval). Where the credible interval includes the known proportion (i.e., model output is accurate), cells are shaded green. P = process error, R = residual error. DIC = Deviance Information Criterion.

[a] Bolded values in parentheses indicate known dietary proportions.

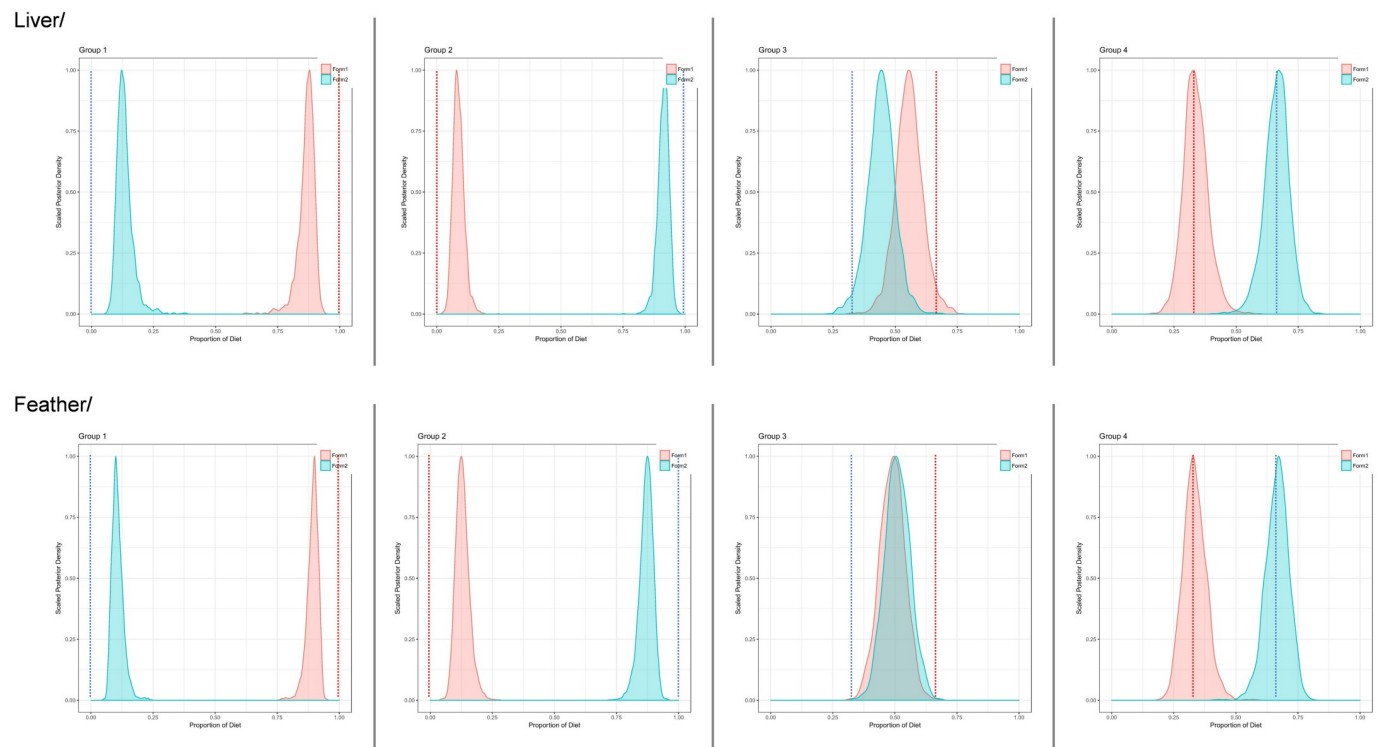

**Fig 5. Estimated contributions (Bayesian credible intervals for scaled posterior densities) of each formula (= 'form', as legend), for both liver and feather.** Estimates based on top-ranked Bayesian stable isotope mixing models were similar to known dietary contributions (indicated by vertical dotted lines), with some variation in accuracy related to dietary group and tissue type.

the opposite. When we included estimates of TDFs that were combined across dietary sources, model performance and accuracy of diet assignments were improved, relative to models with diet-specific estimates. Although this is in apparent contrast to the idea that diet-specific estimates of TDFs are more valuable for isotopic dietary assignments than literature-based values [5, 19], our findings align with work suggesting that larger uncertainties in TDFs can improve fit and accuracy of BSIMMs [24, 75].

Regarding the accuracy of BSIMM outputs under best-case conditions, we note that under both TDF specifications, relative dietary contributions estimated by the Bayesian credible intervals were similar to known proportions (Tables 4 and 5), being within about 10% of the known diet (or better) in all cases we looked at. This is encouraging for use of BSIMMs to identify diets in field studies, albeit not perfect. Importantly, our results, particularly for dietary groups 3 and 4, suggest that dominant dietary components would be clearly identified by BSIMM outputs, even though accuracy of estimates might vary. Ecologists can therefore expect predicted diets based on these models will give solid insights into the dominant assimilated dietary components, at least in food webs with isotopically distinct dietary options. We recommend, however, that results not be considered as indicative of strictly true diets, as revealed in our "moment of truth".

Because our results indicate that accuracy, more than inference, is affected by TDF variance estimates when isotopic differences between source proportions are high, we recommend that future studies should consistently consider overall sensitivities of BSIMM outputs to TDF variance, particularly because assuming single, fixed TDFs can artificially reduce estimates of sampling error and the importance of ecological processes such as individual specialization [16]. That models with process-related error terms (for both tissue types) were more supported by our data than residual error only models is consistent with the idea that variability in consumer isotope values due to chance–as well as due to distinct dietary choices of individual animals–are important elements of mixing models. This was an unexpected finding for our captive setting, where we expected that the relatively homogenous nature of the pelleted feed would prevent individuals from preferentially consuming one formula (i.e., source) over another. We also assumed that by using the same lipid extraction methods for all diet and liver samples (feathers do not contain lipids), possible effects of lipid extraction on BSIMM outputs would be negligible [76, 77]; indeed, model-predicted dietary assignments for lipid-rich liver and lipid-free feather tissues were similar (Fig 5).

Although the importance of process-related error in our models implies that source selectivity among individual consumers may be underestimated in many natural settings, we caution that factors leading to individual specialization are likely dependent on variation in an individual's state, as well as competitive pressure for resources [78, 79]. We suggest that additional studies assessing intraspecific dietary variation are needed to fully understand the extent of this phenomenon and the conditions under which it is expected to arise. We also further echo the suggestions of others that understanding the underlying ecological assumptions of different BSIMM formulations, particularly error structures, is key to improving inference in dietary studies [13, 80, 81].

## Supporting information

**S1 Table. Commercially available feed ingredients were used to formulate experimental diets.** Percentages of each constituent are shown.
(PDF)

## Acknowledgments

We thank Jamille McLeod, Robert Gonda, Steve Leach, Lisha Berzins, and Katelyn Luff for assistance with sample collection and processing, Stuart Bearhop for encouragement, anonymous reviewers for their constructive comments, and the University of Saskatchewan's Poultry Centre and Department of Soil Science for logistics support.

## Author Contributions

**Conceptualization:** Kirsty E. B. Gurney, Henry L. Classen.

**Data curation:** Kirsty E. B. Gurney.

**Formal analysis:** Kirsty E. B. Gurney, Robert G. Clark.

**Investigation:** Kirsty E. B. Gurney, Henry L. Classen, Robert G. Clark.

**Methodology:** Kirsty E. B. Gurney, Henry L. Classen, Robert G. Clark.

**Project administration:** Kirsty E. B. Gurney, Henry L. Classen, Robert G. Clark.

**Writing – original draft:** Kirsty E. B. Gurney, Robert G. Clark.

**Writing – review & editing:** Kirsty E. B. Gurney, Henry L. Classen, Robert G. Clark.

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
