## [Decision Letter · Decision Letter 0]

1 Apr 2024

PONE-D-24-03114Testing for effects of growth rate on isotope trophic discrimination factors and evaluating the performance of Bayesian stable isotope mixing models experimentally: a moment of truth?PLOS ONE

Dear Dr. Gurney,

Thank you for submitting your manuscript to PLOS ONE. I have now received two reviews that judge the work to be of value and suitable for publication. The reviewers have provided some recommendations for improvements that should make the contribution clearer and more statistically rigorous. Consequently, I conclude that the manuscript, while having merit, should be returned to you and your co-authors for revision along fairly limited lines. I invite you to submit a revised version of the manuscript that addresses the points raised during the review process.

**Reviewer #2 in particular recommends that in the revision comparisons of mixing model results should be made across feeding groups, and this reviewer also thought that the reporting of model results in the tables was not as clear as it should be. Please consider the recommendations made by both reviewers as you proceed with your revision.**

We look forward to receiving your revised manuscript.

Kind regards,

Lee W Cooper, Ph.D.

Section Editor

PLOS ONE

Journal Requirements:

Environment and Climate Change Canada

3. Thank you for uploading your study's underlying data set. Unfortunately, the repository you have noted in your Data Availability statement does not qualify as an acceptable data repository according to PLOS's standards.

6. We note that you have referenced Campos D, Macari M, Fernadez-Alarcon M, Nogueira W, de Souza F, Hada F, et al. which has currently not yet been accepted for publication. Please remove this from your References and amend this to state in the body of your manuscript: (Campos D, Macari M, Fernadez-Alarcon M, Nogueira W, de Souza F, Hada F, et al. [Submitted]) as detailed online in our guide for authors

Reviewers' comments:

Reviewer's Responses to Questions

**Comments to the Author**

1. Is the manuscript technically sound, and do the data support the conclusions?

Reviewer #1: Yes

Reviewer #2: Yes

2. Has the statistical analysis been performed appropriately and rigorously? 

Reviewer #1: Yes

Reviewer #2: No

3. Have the authors made all data underlying the findings in their manuscript fully available?

Reviewer #1: Yes

Reviewer #2: Yes

4. Is the manuscript presented in an intelligible fashion and written in standard English?

Reviewer #1: Yes

Reviewer #2: Yes

5. Review Comments to the Author

**Reviewer #1:** Thanks for the opportunity to review your manuscript. I feel that it is a solid piece of work and that relatively minor revisions are needed to make it suitable for publication.

My specific comments are as follows:

Lines 55-56: I don’t think inert tissues are necessarily sampled non-invasively. For example, plucking a feather or clipping a nail is considered invasive sampling.

Lines 184-185: It doesn’t appear that the many of the measured d15N values ended up falling within the range of the values of the standards. What influence might this have on the data?

Lines 170-193: I was expecting this section to also detail how the isotope data were normalized to the international scale, but that was missing. Please add those details.

Lines 378-379: Although there is a statistical effect of sex on TDF for d13C, the size of this effect (which appears to be <0.3 per mil) is quite small, which seems worth saying somewhere in this paragraph.

Discussion: I didn’t feel that the Discussion completely addressed this accuracy-related question that was stated on lines 107-108. Please consider revising the Discussion to do so.

**Reviewer #2: **This study is original and well designed; and the manuscript is well written. I should be acceptable for publication following minor revisions. Specifically, while most of the statistical analyses used in the manuscript is appropriate, I suggest that authors revise their reporting and comparisons of mixing model results across feeding groups. The current approach uses qualitative comparisons and the mean and SD reporting of model results in Tables 4 and 5 makes it challenging for the reader to understand what is being reported in these tables. Specific recommendations are outlined below:

Line 228-229: This section described how models with different terms/structures were ranked using DIC but does not include text on how the results of these model quantitatively were compared across diet groups. Based on Figure 5, it appears that this was done in qualitative manner by plotting the actual diet proportions for each group on mixing model posterior distribution plots. I recommend that the authors formalize and describe their approach to comparing model results here in the methods. Specifically, I would recommend that qualitatively compare the actual diet proportions for each group to the 95% CI resulting from each mixing model. If the actual falls within the 95%CI then it is “similar” (i.e., mixing model is accurate) but if it does not than they are different (i.e., mixing model is not accurate).

Table 4 & 5: I find these tables very hard to interpret even with the sub-caption. Also it is uncommon to report the results of Bayesian mixing models as Mean±SD. Reporting mixing model results as means ± SDs can lead to biased interpretations and fundamentally does not align with the probabilistic concepts underlying the Bayesian statistics used in these models. While others have reported Bayesian mixing model results in this manner it does not mean that doing so is always appropriate. This is because MixSIAR estimates proportions (that must sum to 1) they are almost always skewed (not a normal distribution). In addition, they can also be multimodal (though that does not appear to be the case in this dataset). That said the posterior distribution plots provided in Figure 5 indicate some long tails that are better accounted for with median and 95%CI intervals provided by MixSIAR. Therefore, please report mixing model results as median values with upper and lower 95% credibility intervals in Table 4 and 1. This will also facilitate the more quantitative comparison between diet groups recommended above.

Other comments:

- Avoid “isotopic signatures” and use “isotopic values” instead.

- Line 173: lipid extraction can bias d15N values in tissues, which can in turn bias mixing model results. (Tarroux, A., Ehrich, D., Lecomte, N., Jardine, T. D., Beˆty, J., & Berteaux, D. (2010). Sensitivity of stable isotope mixing models to variation in isotopic ratios: evaluating consequences of lipid extraction. Methods in Ecology and Evolution, 1(3), 231-241.). Please clarify how this might have affected your results and address this in the discussion.

- Line 214: Why 85% and not 95%?

6. PLOS authors have the option to publish the peer review history of their article (what does this mean?). If published, this will include your full peer review and any attached files.

Reviewer #1: No

Reviewer #2: No

---

## [Author Response · Author response to Decision Letter 0]

8 May 2024

Dr. Lee W. Cooper

Section Editor, PLOS ONE

University of Maryland

Solomons, Maryland

May 7, 2024

Dear Dr. Cooper;

We are grateful for an opportunity to submit a revised version of our manuscript, “Testing for effects of growth rate on isotope trophic discrimination factors and evaluating the performance of Bayesian stable isotope mixing models experimentally: a moment of truth?” (PONE-D-24-03114), by K. Gurney et al. The review comments have helped us to improve the clarity and rigour of the manuscript. Based on your most recent correspondence (01 April 2024), we understand that main concerns relate to:

(1) compliance with journal requirements (as below), and 

(2) statistical comparisons of mixing model results.

We discuss each of these issues in detail when responding to each individual comment (in italics, below), but highlights of our responses are also provided here, for summary purposes: 

(1) To ensure that we are in strict compliance with journal formatting and data sharing requirements, we have revised text throughout the document, as requested. We have also uploaded our data to Zenodo and published it, so that it is now publicly available.

(2) We have revised the statistical approach as suggested by Reviewer 2 and have made modifications to the Tables (and captions) that we hope will make them more reader-friendly. 

Journal Requirements: 

• Response: We have carefully reviewed the style templates and made every effort to ensure that the revision is formatted in line with PLOS ONE’s requirements. If further formatting changes are required, we can make those changes quickly.

Environment and Climate Change Canada

If this statement is not correct you must amend it as needed. Please include this amended Role of Funder statement in your cover letter; we will change the online submission form on your behalf.

• Response: Funding was provided by Environment and Climate Change Canada. The funders had no role in study design, data collection and analysis, decision to publish, or preparation of the manuscript. 

3. Thank you for uploading your study's underlying data set. Unfortunately, the repository you have noted in your Data Availability statement does not qualify as an acceptable data repository according to PLOS's standards.

• Response: Thank you for sharing the link to the recommended repositories. We have uploaded our complete data sets to the Zenodo general-purpose open repository. The DOI that can be used to access these data is https://doi.org/10.5281/zenodo.10927544

• Response: As requested, our entire data set is now freely accessible on the Zenodo general-purpose open repository (https://doi.org/10.5281/zenodo.10927544) .

• Response: The full ethics statement is included in the ‘Methods’ section of the revised manuscript (lines 149–151) and has been modified slightly to include the names of both the Institutional Research Board (i.e., Animal Research Ethics) and the Committee (i.e., Animal Care Committee), which provided written approval of the study.

6. We note that you have referenced Campos D, Macari M, Fernadez-Alarcon M, Nogueira W, de Souza F, Hada F, et al. which has currently not yet been accepted for publication. Please remove this from your References and amend this to state in the body of your manuscript: (Campos D, Macari M, Fernadez-Alarcon M, Nogueira W, de Souza F, Hada F, et al. [Submitted]) as detailed online in our guide for authors

• Response: With apologies for my confusion, I am uncertain as to why this article is considered as not yet accepted for publication? I find that it was published in the British Poultry Science journal (January 2016) and that the open access article (DOI: 10.1080/00071668.2015.1115467; ISSN 00071668) is listed both in Web of Science and Scopus.

• If there is further concern with this reference, please advise.

• Response: The reference list has been carefully reviewed and is complete and correct.

Reviewers' comments to the Author

Lines 55-56: I don’t think inert tissues are necessarily sampled non-invasively. For example, plucking a feather or clipping a nail is considered invasive sampling.

• Response: We have modified this sentence to state that the sampling of inert tissues is non-lethal and less invasive than sampling of metabolically active tissues (lines 55–57).

Lines 184-185: It doesn’t appear that the many of the measured d15N values ended up falling within the range of the values of the standards. What influence might this have on the data?

• Response: We agree that it is ideal to have the secondary reference materials bracket sample isotopic compositions, but this can be challenging to achieve in all cases. Studies that have reviewed this issue (listed below) indicate that secondary reference materials that do not bracket the experimental values do not have a statistically meaningful influence on the data, as calibration – especially for carbon and nitrogen – most commonly remains linear beyond the range of the used reference materials. 

o Bond, Alexander L., and Keith A. Hobson. "Reporting stable-isotope ratios in ecology: recommended terminology, guidelines and best practices." Waterbirds 35.2 (2012): 324-331.

o Carter, J. F., and Brian Fry. "Ensuring the reliability of stable isotope ratio data—beyond the principle of identical treatment." Analytical and Bioanalytical Chemistry 405 (2013): 2799-2814.

Lines 170-193: I was expecting this section to also detail how the isotope data were normalized to the international scale, but that was missing. Please add those details.

• Response: Thank you for the suggestion. We have added text (lines 185–186 and lines 189–192), further describing the normalization and calibration methods used by the analytical lab. 

Lines 378-379: Although there is a statistical effect of sex on TDF for d13C, the size of this effect (which appears to be <0.3 per mil) is quite small, which seems worth saying somewhere in this paragraph.

• Response: We note that the effect of sex on TDF was observed for d15N, rather than d13C, but have modified the paragraph to highlight the small effect size for sex (line 399).

Discussion: I didn’t feel that the Discussion completely addressed this accuracy-related question that was stated on lines 107-108. Please consider revising the Discussion to do so.

• Response: Thank you. Based on feedback from Reviewer 2, we have revised the manuscript content to include more information on assessing and describing accuracy (see lines 236–240 and 327–332; Tables 4 and 5).

• We also refer back to these descriptions when discussing model performance, especially lines 435–445, where we note that relative dietary contributions were similar to known proportions. We hope that these adjustments address the Reviewer’s concern.

Reviewer #2: This study is original and well designed; and the manuscript is well written. I should be acceptable for publication following minor revisions. Specifically, while most of the statistical analyses used in the manuscript is appropriate, I suggest that authors revise their reporting and comparisons of mixing model results across feeding groups. The current approach uses qualitative comparisons and the mean and SD reporting of model results in Tables 4 and 5 makes it challenging for the reader to understand what is being reported in these tables. Specific recommendations are outlined below:

Line 228-229: This section described how models with different terms/structures were ranked using DIC but does not include text on how the results of these model quantitatively were compared across diet groups. Based on Figure 5, it appears that this was done in qualitative manner by plotting the actual diet proportions for each group on mixing model posterior distribution plots. I recommend that the authors formalize and describe their approach to comparing model results here in the methods. 

Specifically, I would recommend that qualitatively compare the actual diet proportions for each group to the 95% CI resulting from each mixing model. If the actual falls within the 95%CI then it is “similar” (i.e., mixing model is accurate) but if it does not than they are different (i.e., mixing model is not accurate).

• Response: We are grateful for this reviewer’s suggestions for improving interpretation of model outputs. We now provide 95% credible intervals (and medians) for all model outputs (Tables 4 and 5). We also highlight in these tables where the credible interval includes the known proportions – for the best approximating model (both tissues). 

• To describe these changes, we have added text to the methods (lines 236–239) and have also added more details to highlight our findings in the results (lines 327–332). We also make reference to the revised tables in the Discussion when mentioning that accuracy of the model outputs (rather than inference) is affected more by model specifications (line 435–437).

Table 4 & 5: I find these tables very hard to interpret even with the sub-caption. Also it is uncommon to report the results of Bayesian mixing models as Mean±SD. Reporting mixing model results as means ± SDs can lead to biased interpretations and fundamentally does not align with the probabilistic concepts underlying the Bayesian statistics used in these models. While others have reported Bayesian mixing model results in this manner it does not mean that doing so is always appropriate. This is because MixSIAR estimates proportions (that must sum to 1) they are almost always skewed (not a normal distribution). In addition, they can also be multimodal (though that does not appear to be the case in this dataset). 

That said the posterior distribution plots provided in Figure 5 indicate some long tails that are better accounted for with median and 95%CI intervals provided by MixSIAR. Therefore, please report mixing model results as median values with upper and lower 95% credibility intervals in Table 4 and 5. This will also facilitate the more quantitative comparison between diet groups recommended above.

• Response: Again, we thank the reviewer for this feedback and have made efforts to clarify the tables through improved captions. Also, as indicated previously, both tables now include median and 95% credible intervals for all model outputs. Please see lines 327–332 for the more quantitative comparison between diet groups. 

Other comments:

- Avoid “isotopic signatures” and use “isotopic values” instead.

• Response: Changes (x4) as suggested. Please see lines 73, 102, 114, and 368.

- Line 173: lipid extraction can bias d15N values in tissues, which can in turn bias mixing model results. 

(Tarroux, A., Ehrich, D., Lecomte, N., Jardine, T. D., Bêty, J., & Berteaux, D. (2010). Sensitivity of stable isotope mixing models to variation in isotopic ratios: evaluating consequences of lipid extraction. Methods in Ecology and Evolution, 1(3), 231-241.). 

Please clarify how this might have affected your results and address this in the discussion.

• Response: Although the effects of lipid extraction on estimated dietary proportions by BSIMMs in this study are not known, we note that (i) all samples were treated (i.e. extracted) similarly using established methods, and (ii) all diets had similar C:N ratios, such that effect of lipid extraction on �13C of food samples should have been similar across groups, with minimal expected effect on �15N for most tissues (egg being an exception, which we did not use), as described by Tarroux et al. (see pages 232 and 237).

• Tarroux et al. also find (see page 237) that when lipid extraction of a consumer tissue strongly effects its isotopic values (i.e., higher lipid content in the bulk tissue), not lipid extracting that tissue can lead to incorrect conclusions about diet. For this reason, we consider that extraction of lipids from liver (a relatively lipid-rich tissue) would have been unlikely to invalidate our estimates of diet composition. Feather tissues, conversely, do not contain lipids, so the potential for an effect of using lipid-extracted isotope values for food items on BSIMMs outputs would be higher; however, estimates of accuracy for both liver and feather tissues were similar, suggesting that this was not the case.

- Line 214: Why 85% and not 95%?

• Response: As noted in the cited reference (Arnold 2010, full citation shown below), the use of 85% confidence intervals is more compatible with model selection using information theory, resulting in reduced ambivalence with respect to selecting informative parameters. This rationale is highlighted at lines 221–224.

o Arnold TW. Uninformative parameters and model selection using Akaike's information criterion. Journal of Wildlife Management. 2010;74(6):1175-8.

---

## [Editor Report · Decision Letter 1]

14 May 2024

Testing for effects of growth rate on isotope trophic discrimination factors and evaluating the performance of Bayesian stable isotope mixing models experimentally: a moment of truth?

PONE-D-24-03114R1

Dear Dr. Gurney,

Thank you for submitting your revised manuscript with attention to meeting or responding to the concerns of the editorial office, as well as the two reviewers who made suggestions that were helpful in improving the presentation of the data and the associated statistics. As a result, I am pleased to inform you that your manuscript has been judged scientifically suitable for publication and will be formally accepted for publication following a review of any last technical requirements by the editorial office.

Again, thank you for choosing PLOS ONE for presentation of your research results, and I join you in looking forward to the publication of your manuscript.

Kind regards,

Lee W Cooper, Ph.D.

Section Editor

PLOS ONE